# Oncology professionals' perspectives towards cardiac surveillance in breast cancer patients with high cardiotoxicity risk: A qualitative study

**Yvonne Koop**[1]*, **Laura Dobbe**[2], **Angela H. E. M. Maas**[1], **Dick Johan van Spronsen**[3], **Femke Atsma**[4], **Saloua El Messaoudi**[1], **Hester Vermeulen**[4,5]

**1** Department of Cardiology, Radboud University Medical Center, Nijmegen, The Netherlands, **2** Department of Medical Oncology, Radboud University Medical Center, Nijmegen, The Netherlands, **3** Department of Hematology, Radboud University Nijmegen Medical Center, Nijmegen, Netherlands, **4** Scientific Institute for Quality of Healthcare, Radboud University Medical Center, Nijmegen, The Netherlands, **5** Research Department of Emergency and Critical Care, Faculty of Health and Social Studies, HAN University of Applied Sciences, Nijmegen, The Netherlands

* Yvonne.Koop@radboudumc.nl

**Data Availability Statement:** Data cannot be shared publicly because local privacy regulations prohibit data sharing of potentially identifying

## Abstract

Breast cancer (BC) patients have an increased risk of developing cancer therapy-related cardiac dysfunction (CTRCD) and cardiovascular morbidity, which seems to have a substantial prognostic impact. Oncologists, in collaboration with dedicated cardiologists, have the opportunity to perform cardiovascular risk stratification. Despite guideline recommendations, strategies to detect cardiac damage at an early stage are not structurally implemented in clinical practice. The perspectives of oncology professionals regarding cardiac surveillance in BC patients have not been qualitatively evaluated. We aim to explore the perceptions of oncology professionals regarding cardiac surveillance in BC patients and, more specifically, the influencing factors of delivering cardiac surveillance. A qualitative study with semi-structured interviews was conducted and thematically analyzed. Twelve oncology professionals participated in this study. Four themes were selected to answer the study objectives: (1) sense of urgency, (2) multidisciplinary collaboration, (3) patient burden, and (4) practical tools for cardiac surveillance. Most professionals did not feel the need to deliver cardiac surveillance as they considered the incidence of CTRCD as rare. Multidisciplinary collaboration was also perceived as unnecessary, and cardiac surveillance was considered disproportionately burdensome with respect to its benefits. Nevertheless, professionals affirmed the need for practical tools to deliver cardiac surveillance. Most professionals are currently unaware of CTRCD incidence and cardiac surveillance benefits. Encouraging multidisciplinary collaboration and improving their knowledge of cardiotoxic effects of treatments and possibility of early detection can lead to structured cardiac surveillance for breast cancer patients.

information, therefore public data sharing was not a part of the written informed consent obtained from the participants. Data access can be requested via the Radboud Research Information Service (contact via ris@ubn.ru.nl) for researchers who meet the criteria for access to confidential data.

**Funding:** The authors received no specific funding for this work.

**Competing interests:** The authors have declared that no competing interests exist.

**Abbreviations:** BC, Breast cancer; CMR, Cardiovascular magnetic resonance; COREQ, Consolidated criteria for reporting qualitative studies; CTRCD, Cancer therapy-related cardiac dysfunction; ESC, European society of Cardiology; ESMO, European society for medical Oncology; LVEF, Left ventricular ejection fraction; MUGA, Multigated acquisition scan; TICD, Tailored implementation for chronic diseases.

# Introduction

Cancer therapy-related cardiac dysfunction (CTRCD) has an incidence of approximately 30% in patients receiving anthracycline-based chemotherapy and trastuzumab, especially after prior radiation therapy [1, 2]. Development of cardiovascular morbidity significantly affects the prognosis of breast cancer (BC) patients (e.g., 5-year heart failure mortality is >50%; [3–6]). There is increasing evidence for the benefits of detecting CTRCD and treating patients with early signs of myocardial damage even before the left ventricular ejection fraction (LVEF) is reduced [7, 8]. Increased interest in cardiac surveillance in cancer patients resulted in the European Society for Medical Oncology's (ESMO) formulation of guidelines and the European Society of Cardiology's (ESC) position paper that presented recommendations for current practice [3, 9, 10]. For patients with a high risk of developing cardiovascular toxicity (e.g., those receiving anthracyclines or trastuzumab), baseline and 3-month monitoring are recommended with cardiac imaging modalities, such as echocardiography or cardiovascular magnetic resonance (CMR), in combination with assessment of cardiac biomarkers [3, 9, 10].

These recommendations have been adopted in the Dutch national guidelines [11]. Recent studies have shown that these recommendations are not yet implemented in clinical practice [12], and several explanations for this discrepancy have been discussed. Physician factors–and not necessarily patient factors or factors related to the proposed anti-cancer therapy–seem crucial in clinical decision making regarding cardiac surveillance in cancer patients [13]. Knowledge of potential barriers for cardiac surveillance could provide insights and objectives for improvement. Therefore, in this study, we aim to qualitatively explore the perceptions of oncology professionals regarding cardiac surveillance in BC patients and, more specifically, the influencing factors of delivering cardiac surveillance.

# Methods

## Design

We conducted a qualitative study with semi-structured interviews to explore oncology professionals' views and expectations regarding cardiac surveillance in BC patients with a high cardiovascular risk. The consolidated criteria for reporting qualitative studies (COREQ) checklist was used to ensure complete reporting of methodology. A waiver was provided by the Medical Research Ethics Committee of Arnhem-Nijmegen because the study did not need an ethical review. The study was conducted in accordance with the principles of ICH Good Clinical Practice, applicable privacy requirements, and principles of the Declaration of Helsinki.

## Participants and recruitment

A purposive sample with maximum variation in gender, work experience, and hospital type was selected with the aim to reflect the true variation in characteristics observed in clinical practice. Twelve oncology professionals specialized in BC care participated in this study: oncologists ($N = 6$), a cancer epidemiologist ($N = 1$), and oncology nurse practitioners ($N = 5$). In this paper, oncology professionals are referred to as professionals, unless otherwise specified. The professionals worked at the medical oncology department in either a university or general hospital. They were selected after consulting the head of the respective oncology departments. Eligibility was based on their experience in BC care: a minimum of 1 year in their current function. The coordinating investigator contacted eligible professionals ($N = 16$) from seven different hospitals via phone and invited them to participate in this qualitative study. Twelve professionals (75%) agreed to participate. Four professionals were unwilling to

participate: one did not have any experience with cardiovascular diseases in BC patients, whereas the other three declined owing to logistic reasons.

## Data collection

A semi-structured interview guide was developed, theoretically based on the seven domains of the Tailored Implementation for Chronic Diseases (TICD) checklist (Table 1), which is a synthesis of 12 existing checklists for identifying determinants of practice and factors that prevent or enable improvements in healthcare professional practice [14].

The interview guide was repeatedly reviewed by all investigators to ensure the feasibility and completeness of the topics; it consisted of open questions regarding clinical protocols, awareness, multi-disciplinary collaboration, organizational structure, and resources (S1 File). All interviews started with the same opening question about clinical protocols for cardiac surveillance in BC patients. All seven TICD domains were discussed. During the interviews, summaries, probes and prompts were used to ensure correct interpretation of professionals' perceptions and encourage outspokenness [15].

Individual face-to-face interviews were conducted in a private room at the hospital where the participant was employed. Baseline characteristics were collected after informed consent was obtained and before the interview began. The interviews were conducted by a 35-year-old, independent, female health sciences student (MSc) with 2 years of work experience in the field of oncology (author LD). The interviewer was unknown to the professionals, which allowed

**Table 1. Interview topics.**

| TICD checklist domain | Determinants |
|---|---|
| Guideline factors | Quality and clarity of recommendations |
| | Accessibility and source of recommendations |
| | Recommended clinical intervention feasibility and accessibility |
| | Compatibility of recommended behavior |
| | Practice with recommendations |
| Individual health professional factors | Knowledge and skills |
| | Attitude and understanding towards recommendations |
| | Professional behavior |
| Patient factors | Patient needs, knowledge and preferences |
| Professional interactions | Communication and influence |
| | Team processes |
| | Referral processes |
| Incentives and resources | Availability of necessary resources |
| | (Non)financial incentives |
| | Information system |
| | Quality assurance systems and assistance for adherence |
| Capacity for organizational change | Authority, accountability |
| | Leadership |
| | Regulations, policies |
| | Priority of change |
| Social, political and legal factors | Individual influence |
| | Contracts |
| | Funding policies |

TICD: Tailored implementation for chronic diseases [14].

them to express their experiences and perceptions without inhibitions. All interviews were conducted in Dutch, all quotes were translated to English with the forward backward procedure for translation reliability.

The interview duration ranged from 21 to 64 min (median: 36 min). The interviews were audio-recorded, transcribed verbatim, and checked for transcription errors. Field notes were made immediately after the interviews to record observations, capture initial ideas on the topics, and reflect on the methodology (e.g., interview guide refinement). Interview techniques were practiced with a simulation interview to enhance the interview skills and obtain feedback from more experienced researchers.

## Data analysis

Thematic analysis as outlined by Braun and Clarke was used, an inductive, semantic and realist approach was applied. Thematic analysis consists of six steps: familiarizing with the data, generating initial codes, searching for themes, reviewing themes, defining and naming themes, and writing (Table 2; [16]). The supplementary 15-point checklist ensured the correct use of the thematic analysis method.

The first four interviews were coded by three researchers independently of each other and subsequently checked for inconsistencies and similarities; the remaining interviews were coded by two independent researchers. Inconsistencies were discussed in analysis meetings with the research team. After the first three interviews, the analysis step "searching for themes" was performed. After the remaining interviews, each theme was further considered and refined; illustrative quotes were also selected. Data saturation was defined as the point where additional interviews did not lead to any codes that introduced a new topic, which was reached after ten interviews. Two more interviews confirmed saturation and ensured maximum variation in the sample.

After transcribing the interviews, 740 codes were generated and grouped in 32 clusters, such as multidisciplinary collaboration, continuity of care, burden, need for knowledge, urgency, logistics, awareness, personalized care, research, visibility of late cardiac effects, and

**Table 2. Six steps of thematic analysis.**

| Phase | Description of the process |
|---|---|
| 1. Familiarizing with the data | Interviews were transcribed (LD); the transcripts were read and re-read; and initial ideas for topics were discussed in the research team (LD, YK). |
| 2. Generating initial codes | All transcripts were coded by two researchers independently of each other (LD, NvZ, YK). Noteworthy features of the data were coded in a systematic fashion across the entire data set, collating data relevant to each code. The codes were presented and discussed in the research team (LD, YK, HV). |
| 3. Searching for themes | Codes were collated into potential themes, gathering all data relevant to each potential theme. A preliminary description of potential themes and subthemes was made and discussed (LD, YK, HV). |
| 4. Reviewing themes | The preliminary themes were checked for consistency with the original data (LD, YK). |
| | Inconsistencies were discussed, and the themes were further explored (LD, YK). The main themes and subthemes were revised accordingly and further described (LD, YK) and reviewed (YK, HV, SEM, DJS, AHEMM, FA). |
| 5. Defining and naming themes | The specifics of each theme were discussed, and names and definitions of themes were refined (YK, LD, HV, SEM, DJS, AHEMM, FA). |
| 6. Producing the report | A first draft of the results was written (YK) and reviewed (HV, SEM, DJS, AHEMM, FA). |
| | The quotes were selected to clarify the presented data; the report was further discussed (LD, YK, SEM, HV) and adjusted (YK). The report was critically assessed by the research team and further modified to adequately present the themes with verbatim quotes (YK, LD, HV, SEM, DJS, AHEMM, FA). |

cost effectiveness. The recurring themes within these clusters were discussed to identify the final themes to answer the research question.

Regular data collection and analysis meetings were scheduled with the interviewer and experienced researchers, which resulted in researcher triangulation and peer review in all phases of the study; this enhanced the methodological quality, reliability, data dependability, and accuracy [15, 17, 18]. Additionally, an audit trail improved the confirmability of the study [15]. Data management was supported by ATLAS.ti software (version 8; Scientific Software Development GmbH, Berlin, Germany).

## Results

Between January and May 2019, 12 oncology professionals in BC care from both general and university hospitals in the Netherlands were interviewed. The mean age of the participants was 52.5 years (range: 36–62 years). Two of the 12 participants were male (17%), and 10 were female (83%). Six of the professionals were oncologists (50%), five were nurse practitioners (42%), and one was an epidemiologist (8%). Most professionals had a master's degree ($N = 7$; 58%) and had >6 years of work experience in their current function ($N = 10$; 83%). Eight professionals (67%) worked at a general hospital, whereas four (33%) worked at a university hospital. The participating professionals were employed at seven different hospitals. Maximum variation was achieved for age and work experience. The baseline characteristics of the study participants are presented in Table 3.

The following sections describe current practice based on the explanations given by the professionals. The four themes that answer the study objectives are also discussed: (1) sense of urgency, (2) multidisciplinary collaboration, (3) patient burden, and (4) practical tools for cardiac surveillance (Fig 1).

### Current practice

Most oncology professionals ($N = 10$; 83%) stated that cardiac surveillance for BC patients is not structurally performed in their hospital. The majority of professionals ($N = 10$ [83%]; from 5 out of 7 [71%] hospitals) disclosed that cardiac monitoring focused on the performance of a multigated acquisition (MUGA) scan for the assessment of the LVEF prior to and during

**Table 3. Baseline characteristics of participants.**

| Participant | Age * (years) | Function | Sex | Highest level of education | Work experience (years) * | Work setting ** |
|---|---|---|---|---|---|---|
| 1 | 54 | Nurse practitioner | Female | Master | 5 | University hospital |
| 2 | 58 | Nurse practitioner | Female | Master | 18 | Non-university hospital |
| 3 | 54 | Oncologist | Female | PhD | 16 | Non-university hospital |
| 4 | 59 | Nurse practitioner | Female | Master | 7 | Non-university hospital |
| 5 | 51 | Nurse practitioner | Male | Master | 11 | Non-university hospital |
| 6 | 39 | Oncologist | Female | PhD | 7 | University hospital |
| 7 | 59 | Oncologist | Female | PhD | 20 | Non-university hospital |
| 8 | 62 | Epidemiologist | Female | PhD | 36 | Non-university hospital |
| 9 | 51 | Nurse practitioner | Female | Master | 9 | Non-university hospital |
| 10 | 42 | Oncologist | Female | PhD | 7 | University hospital |
| 11 | 41 | Oncologist | Female | Master | 8 | University hospital |
| 12 | 36 | Oncologist | Male | Master | 3 | Non-university hospital |

* At time of interview;

** Non-university hospital: General hospital or top clinical teaching hospital.

# Professionals' perception of cardiac surveillance

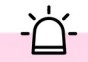

**Sense of urgency**

- No recognition of advantages as CTRCD
  is assumed to be rare
- Lack of knowledge and experience
- Oncologists assume long waiting lists for
  cardiac surveillance
- Cardiologists assume low cancer survival rates

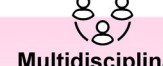

**Multidisciplinary collaboration**

- Essential for holistic care approach
- Facilitates communication
- Research projects to initiate collaboration
- Multidisciplinary collaboration not yet
  implemented in guidelines and clinical practice

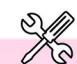

**Practical tools for cardiac surveillance**

- Lack of knowledge in clinical practice
- Protocols are needed for risk stratification
- Practical tools to tailor cardiac surveillance
  are required

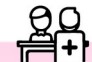

**Patient burden**

- Information on potential side effects is
  perceived as an unnecessary burden
- Assumption that cancer patients are not
  concerned about their cardiovascular health
- Quality of life is an important topic to discuss
- Continuity of care is essential

**Fig 1. Thematic overview.**

trastuzumab treatment for HER2$^+$ BC patients. Patients receiving other (high-risk) cardiotoxic treatments did not receive any form of cardiac surveillance unless it was clinically indicated owing to the development of cardiac symptoms or patients already received it owing to pre-existing cardiovascular comorbidities. Only two professionals from two different hospitals mentioned the recent implementation of a cardio-oncology referral department for cancer patients receiving high-risk cardiotoxic treatment. At these dedicated cardio-oncology outpatient clinics, cardiovascular risk stratification and cardiotoxicity monitoring during cancer treatment are performed. Notably, all professionals indicated that strict clinical protocols (based on guidelines) for the detection of cardiotoxicity during and after cancer treatment are lacking.

## Sense of urgency

All professionals were aware of the cardiotoxic effects of anthracycline- and trastuzumab-based regimens. They all clarified that their delivery of cardiac surveillance was mainly driven by their sense of (or lack of) urgency; they did not perform any form of cardiac surveillance mostly because they did not recognize the advantages.

The assumed low incidence of cardiac events in their patient population was the main cause and played a key role in the professionals' clinical decision making. Oncologists indicated that they rarely treated BC patients with CTRCD; therefore, they perceived the occurrence of (adverse) cardiac events during cancer treatment as rare. Additionally, a reduction in LVEF was considered as reversible and therefore not important to detect using cardiac imaging.

*"I can't remember anyone from recent years who I referred to a cardiologist, no one who expressed complaints that could be of a cardiac nature and then came back with heart failure. And that could mean two things: either those patients weren't there, or we haven't searched properly." (P6)*

Most professionals did not feel the need to change current practice regarding cardiac surveillance despite the increasing awareness on cardiotoxicity among specialists.

*"I remember a patient–she really had heart failure. The question is, if we had done things differently and screened the patient before cancer treatment, we probably couldn't have prevented it anyway. So, in that case, I believe we're still doing everything as we should, and we don't have to change anything, but yes, that one patient–if I may say–was very unlucky. It did have a large impact; yes." (P12)*

Most professionals were not convinced that increased cardiac monitoring can prevent CTRCD or improve quality of life. Those in favor of cardiac monitoring mentioned their colleagues' lack of knowledge on the topic as a barrier. Similarly, the perception of oncologists that cardiology waiting lists are usually long prevented them from referring their patients for cardiac screening because they feared this would significantly delay cancer treatment. Some oncologists disclosed that cardiac monitoring in cancer patients was not a priority for cardiologists because of anticipated poor cancer prognosis.

*"When we say, 'cancer patient', the patient's prognosis is not always correctly interpreted by cardiologists. It quickly sounds like, 'It's cancer. We don't have to proceed with monitoring quickly because the prognosis is already limited.' But that's exactly the question. Obviously, this is true sometimes, but there are also situations in which we expect patients to have many more years to go; I feel [that] it sometimes influences a cardiologist's advice too much." (P10)*

All professionals agreed that increased knowledge of CTRCD, improved awareness of all involved disciplines and practitioners, and more proactive and preventive behavior are important. Knowledge, experience, or patient stories may result in an increased sense of urgency of professionals regarding cardiac surveillance.

*"And then, I had a patient with severe cardiac damage, and she told me, 'It was completely not recognized at all, my cardiac complaints.' She was in tears. Now, I recognize how important it is to properly inquire about potential [cardiac] complaints." (P4)*

## Multidisciplinary collaboration

Multidisciplinary collaboration between oncologists and dedicated cardiologists highly influenced the manner in which cardiac surveillance was organized. Two oncologists involved in multidisciplinary cardio-oncology collaboration acknowledged that this facilitated communication and resulted in more structured cardiac surveillance in BC patients.

*"For us, the logistics are now very easy; we have one contact person, someone who evaluates the risks of a patient [. . .]. Sometimes we felt MUGA results were not sensitive enough. Now [with echo], we know more and are able to make well-thought-out decisions [. . .]. Follow-up is every three months unless it is indicated to schedule it earlier." (P11)*

Multidisciplinary collaborations were often initiated after combined research projects and were implemented by dedicated professionals, both in oncology and cardiology, who have intrinsic motivation and a mutual interest in improving (cardiovascular) clinical outcomes of cancer patients.

*"I see that most things are only achieved when a few people start collaborating. That is the way it works. It often starts with research collaborations." (P8)*

All professionals perceived multidisciplinary cardio-oncology collaboration as an important asset in delivering holistic care that takes into account patients' mental, physical, and social well-being.

*"I am always very medically involved, and the nurse, in my opinion, sees the patients and their [social] system. And I think that the nurse can offer a whole package, more than I can." (P11)*

Professionals without an established cardio-oncology department ($N$ = 10; 83%) revealed that they refer their BC patients to a cardiologist only when the patients have a history of cardiac events. Many of these professionals perceived occasional consultation with a cardiologist as sufficient, given that they considered CTRCD as a rare complication. As such, they believed that multidisciplinary collaboration is not necessary nor recommended in the literature.

*"There is no standard collaboration; [it occurs] only in acute situations. It's just not a frequent complication. But then again, with that last patient [with cardiac damage], well, sometimes not all specialists completely understand or know what kind of treatment we use. I think it's too easy to assume that cardiac damage during treatment is related to that treatment. It's sometimes not clear [. . .]. There is not enough evidence to initiate something or to structurally collaborate. At least we haven't done that so far, and it's also not in the guidelines." (P7)*

### Patient burden

Professionals considered the occurrence of CTRCD as rare and therefore choose to limit informing patients regarding cardiotoxicity risk in an effort to "unburden" them. They stated that they want to avoid overwhelming their patients with extensive information on possible cardiac side effects before the start of essential or even life-saving cancer treatment. Moreover, they felt conflicted in disclosing complications they perceived as rare. Professionals also believed that patients are not very concerned about their cardiovascular health when first diagnosed with BC and also not during or after cancer treatment because staying alive is assumed to be considered of higher importance after a cancer diagnosis.

*"It was just never a reason to give a different treatment. You just explain it. Patients are not very concerned with it [potential cardiac damage]. They are more like, 'Do I need to go to a cardiologist? Ok then' but never concerned about their cardiac function. But, of course, we don't know what they're thinking when they're at home." (P11)*

Nonetheless, all professionals emphasized that quality of life after cancer treatment is an important topic to discuss with patients. They explained that long-term effects should be considered more often during patient counselling.

*"Actually, I think we all want the same thing, which is keeping patients alive. But especially with a good quality of life. We don't do that if we cure patients of cancer but then they die because of heart failure. If that's the case, we did something drastically wrong or, at least, not as we'd wish. In that case, we should learn from this and do something about it." (P6)*

Continuity of care, both during BC treatment and during follow-up, was considered essential to unburden patients. Patients being assessed by different cardiologists during consecutive consultations was perceived by professionals as a negative aspect of cardiac care. Most professionals did not feel the need for patients to regularly visit different healthcare specialists involved in the care process (e.g., surgeon, radiotherapist, and cardiologist). Specifically, nurse practitioners underlined the importance of a familiar face for patients, and they recommended healthcare coordination via one professional to improve healthcare quality and accelerate decision making.

*"Patients were seen by so many specialists, and then they returned to me. And I was like, 'What was the additional value of all these separate visits?' [. . .] It's good to get to know*

*patients, coordinate their trajectory, monitor [them] more easily, and notice when something is going wrong [. . .]. It's good for the continuity we offer if I'm the main contact, then you're more like 'When was the last echocardiography? We need to request a new evaluation.'" (P5)*

Nurse practitioners stated that continuity of care for BC patients has added value and perceived their involvement in the care process, from start to finish, as a strength. Continuity supports nurse practitioners in delivering personalized care. Health changes can be more easily detected, and health education can be tailored to patients' needs.

*"That is the advantage of knowing people, that you don't just check things off your list. But you're more like, 'Who do I have in front of me and to what extent does each aspect [of complaints and health] need attention?'" (P5)*

### Practical tools for cardiac surveillance

All professionals acknowledged the need for practical tools in delivering cardiac surveillance for BC patients. They also underlined the need for clinical protocols and guidelines for cardiac surveillance, with a focus on risk stratification, to distinguish between high- and low-cardiovascular-risk patients and tailor cardiac surveillance accordingly. Oncologists stated that it is difficult to recognize CTRCD without proper knowledge.

*"It would be great if we knew, for breast cancer, if we could identify when someone has a reduction in LVEF during treatment or shows certain specific cardiac complaints or has a risk profile. That you would know that this is a high-risk patient I need to follow up for the next four to five years." (P11)*

Most professionals expressed the need for evidence-based guidelines for cardiac surveillance and their structured implementation in current clinical practice.

*"I really think you need to make a guideline for this [cardiac surveillance]. Not everyone should just start doing things. We need to prevent an abundance of initiatives without proper support from a guideline." (P8)*

Oncologists disclosed a knowledge gap regarding the necessity for cardiac surveillance, which is mainly due to a lack of data on long-term benefits and cost effectiveness.

*"There is always a number-needed-to-harm and a consideration on how many patients are burdened by failed medical activities versus the benefits. That is an important question." (P3)*

All professionals shared the opinion that changes in healthcare practice should be evidence based and require an initiative from professionals themselves.

## Discussion

In this qualitative study, we explored oncology professionals' perspectives regarding cardiac surveillance in BC patients. Most professionals included in our study did not feel the need to deliver cardiac surveillance as they perceived the incidence of CTRCD in BC patients as low. In addition, most professionals believed that a decline in LVEF during cancer treatment is either spontaneously reversible without pharmacological interference or persistent and refractory to heart failure therapy. Burdening patients with information on the potential cardiotoxic effects

of cancer treatment, performing cardiovascular risk stratification, or initiating cardiac monitoring were therefore perceived as logistically challenging, unnecessary, and even disproportionate with respect to potential benefits. Nevertheless, there was increased awareness on cardiotoxicity risks and potential advantages of cardiac monitoring for certain cancer patients. The oncology professionals highlighted a knowledge gap concerning this topic and emphasized the need for strict guidelines to ensure the delivery of tailored cardiac surveillance in BC care.

Contrary to the perception of oncology professionals in our study, cardiotoxicity is a common side effect of BC treatments, with up to 30% of patients developing heart failure after high-risk cardiotoxic treatments (e.g., anthracyclines and trastuzumab; [1–3]). Cardiovascular disease mortality risk even exceeds cancer mortality risk in this population [19–21]. Chemotherapy-induced cardiomyopathy was traditionally considered to have a poor prognosis and was often refractory to heart failure treatment. Recent studies, however, have suggested that a reduction in LVEF can be mitigated when cardiotoxicity is detected at an early stage and timely intervention is provided [7, 8]. Notably, almost all anthracycline-induced cardiotoxicity (98%) can be detected in the first year after treatment [22].

Measurement of global longitudinal strain using echocardiography is a reproducible technique to detect early signs of cardiac dysfunction [23, 24]. Additionally, myocardial edema and diffuse fibrosis present even in early stages of CTRCD can be detected with CMR mapping sequences [25, 26]. Although recent guidelines (national and international) indicate serial monitoring with cardiac imaging in patients receiving high-risk cardiotoxic cancer treatments [9–11], these guidelines are not implemented in clinical care, and healthcare professionals seem to be unaware of their necessity [12].

This finding regarding low awareness among professionals is consistent with results of a qualitative study on BC patients conducted by our research group: Patients stated that their treating professionals were unaware of cardiac side effects of cancer regimens and that their cardiac events were often not timely recognized [27]. Physician factors, and not necessarily patient factors or factors related to cardiotoxicity of anti-cancer therapy, are crucial in clinical decision making regarding cardiac surveillance in cancer patients [13]; therefore, increasing awareness among healthcare professionals is important. Oncology professionals perceive cardiac monitoring as indicated when their patients have already developed symptoms and signs of cardiac dysfunction [28]. At this stage, cardiac damage may be irreversible even though this could have been prevented with surveillance regimens at an earlier stage [7, 8].

Evidence-based specific guidelines are needed to ensure effective clinical protocols and establish partnerships between nurse practitioners and physicians in the oncology, cardiology, and radiology departments. This collaboration can promote knowledge and shared responsibility among professionals regarding cancer therapeutics, cardiotoxic effects, early cardiac damage detection strategies, and cardioprotective treatments [29–31]. Moreover, guidelines should state the survival and quality-of-life benefits of surveillance as well as cost effectiveness to enhance professionals' sense of urgency. A multidisciplinary collaboration can also improve the prognosis and quality of life of BC patients. For survivors of childhood malignancies, structured cardiac surveillance, often organized in dedicated multidisciplinary outpatient clinics, improve outcomes on morbidity and frequency of inpatient admissions [32].

In conclusion, cardiac surveillance and early detection of CTRCD can improve prognosis in BC patients [33]. Professionals seem to underestimate the need for cardiac surveillance; they stress the need for and importance of clinical tools and clear guidelines. Awareness on the importance of cardiac surveillance for cancer patients should be increased, and dedicated multidisciplinary cardio-oncology outpatient clinics can help realize this goal. As cardio-oncology is a relatively young clinical field, future research has to close the knowledge gaps concerning the long-term benefits of cardiac surveillance and optimal surveillance strategies.

### Implications for practice

- Knowledge of cardiotoxic effects and possibility of early detection is essential to reduce long-term cardiac damage.

- Oncology professionals express a need for structured clinical protocols to provide cardiac surveillance based on recommendations for all high-risk BC patients.

- Multidisciplinary collaboration between the cardiology and oncology departments is important to structure and coordinate cardiac surveillance for BC patients, and collaborative research projects can catalyze this process.

### Supporting information

**S1 Table. Interview guide healthcare professionals.** A table is presented with the interview guide; all main questions as well as potential follow-up questions are listed. The order of the questions is based on the seven domains of the TICD checklist, which was the theoretical basis of the interview guide.
(PDF)

### Acknowledgments

We would like to thank all professionals who participated in this study.

### Author Contributions

**Conceptualization:** Yvonne Koop, Angela H. E. M. Maas, Femke Atsma, Saloua El Messaoudi, Hester Vermeulen.

**Data curation:** Yvonne Koop, Laura Dobbe.

**Formal analysis:** Yvonne Koop, Laura Dobbe, Hester Vermeulen.

**Investigation:** Yvonne Koop, Laura Dobbe.

**Methodology:** Yvonne Koop, Angela H. E. M. Maas, Femke Atsma, Saloua El Messaoudi, Hester Vermeulen.

**Project administration:** Yvonne Koop.

**Resources:** Yvonne Koop.

**Software:** Yvonne Koop, Laura Dobbe, Hester Vermeulen.

**Supervision:** Yvonne Koop, Angela H. E. M. Maas, Dick Johan van Spronsen, Femke Atsma, Saloua El Messaoudi, Hester Vermeulen.

**Validation:** Yvonne Koop, Dick Johan van Spronsen, Femke Atsma, Saloua El Messaoudi, Hester Vermeulen.

**Visualization:** Yvonne Koop, Saloua El Messaoudi.

**Writing – original draft:** Yvonne Koop.

**Writing – review & editing:** Yvonne Koop, Angela H. E. M. Maas, Dick Johan van Spronsen, Femke Atsma, Saloua El Messaoudi, Hester Vermeulen.

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
