## [Decision Letter · Decision Letter 0]

18 Feb 2021

PONE-D-20-31818

Oncology professionals’ perspectives towards cardiac surveillance in breast cancer patients with high cardiotoxicity risk: A qualitative study

PLOS ONE

Dear Dr. Koop,

Thank you for submitting your manuscript to PLOS ONE. After careful consideration, we feel that it has merit but does not fully meet PLOS ONE’s publication criteria as it currently stands. Therefore, we invite you to submit a revised version of the manuscript that addresses the points raised during the review process.

The only thing that is a bit  of a concern is the size of your study population.  It is possible to increase?

We look forward to receiving your revised manuscript.

Kind regards,

Katriina Aalto-Setala, Professor

Academic Editor

PLOS ONE

Journal Requirements:

2. In the methods section, please specify which language the interviews were conducted in, and if necessary, how the transcripts were translated for analysis.

Reviewers' comments:

Reviewer's Responses to Questions

**Comments to the Author**

1. Is the manuscript technically sound, and do the data support the conclusions?

Reviewer #1: Yes

Reviewer #2: Yes

2. Has the statistical analysis been performed appropriately and rigorously? 

Reviewer #1: Yes

Reviewer #2: N/A

3. Have the authors made all data underlying the findings in their manuscript fully available?

Reviewer #1: Yes

Reviewer #2: Yes

4. Is the manuscript presented in an intelligible fashion and written in standard English?

Reviewer #1: Yes

Reviewer #2: Yes

5. Review Comments to the Author

Reviewer #1: This is an important study that investigates attitudes of physicians towards cardiac screening. If detected early, cardiac dysfunction can be treated. The study is well constructed and thoughtful.

Concerns:

Major - only 12 physicians were interviewed - and only 2 male. This really limits the impact of the study. At least 30 should be surveyed.

The manuscript should be edited by someone who is a native speaker of English.

Reviewer #2: Manuscript Number: PONE-D-20-31818

Title: Oncology professionals’ perspectives towards cardiac surveillance in breast cancer

patients with high cardiotoxicity risk: A qualitative study

Reviewer comments

This qualitative study about medical oncologist perception regarding cardiac surveillance in breast cancer patients is well written and emphasizes the necessity of guidelines for cardiovascular surveillance after breast cancer treatment.

Only comment is regarding the monodisciplinary approach. The manuscript can be strengthened by expanding the group of interviews to cardiologists/specialists in vascular medicine and radiation oncologists.

It is not clear why an epidemiologist was included in an interview about clinical practices.

6. PLOS authors have the option to publish the peer review history of their article (what does this mean?). If published, this will include your full peer review and any attached files.

Reviewer #1: **Yes: **Victoria Seewaldt

Reviewer #2: No

---

## [Author Response · Author response to Decision Letter 0]

9 Mar 2021

Comment 1 reviewer 1

Only 12 physicians were interviewed - and only 2 males. This really limits the impact of the study. At least 30 should be surveyed.

Response

We thank the reviewer for this comment. We conducted interviews up to the point of data saturation. After the tenth interview, data saturation was reached, which means that no new codes (i.e., no codes that introduced a new topic) were added to the data. We conducted two more interviews to confirm the data saturation. After data saturation was reached, it was deemed methodologically unnecessary to conduct additional interviews because sufficient data had been collected to answer the specific research question. 

We have revised the methods and results sections to clarify this concern.

Methods (P 6, Ln 11-18):

Data saturation was defined as the point where additional interviews did not lead to any codes that introduced a new topic, which was reached after ten interviews. Two more interviews confirmed saturation and ensured maximum variation in the sample.

After transcribing the interviews, 740 codes were generated and grouped in 32 clusters, such as multidisciplinary collaboration, continuity of care, burden, need for knowledge, urgency, logistics, awareness, personalized care, research, visibility of late cardiac effects, and cost effectiveness. The recurring themes within these clusters were discussed to identify the final themes to answer the research question.

The limited variation in gender of the interviewees is a reflection of the gender distribution of breast cancer professionals in general. Most oncology professionals with a specific focus on breast cancer patients are female. 

To clarify these considerations, we revised the methods section (P 4, Ln 12-13):

A purposive sample with maximum variation in gender, work experience, and hospital type was selected with the aim to reflect the true variation in characteristics observed in clinical practice.

Comment 2 reviewer 1

The manuscript should be edited by someone who is a native speaker of English.

Response 

The manuscript had been checked by a native speaker before submission to PLOS ONE. Following reviewers’ comments, we asked a second native speaker to edit the manuscript, and any remaining grammatical errors were corrected.

Comment 1 reviewer 2

Only comment is regarding the monodisciplinary approach. The manuscript can be strengthened by expanding the group of interviews to cardiologists/specialists in vascular medicine and radiation oncologists.

Response

We thank the reviewer for this comment, we agree that professionals from cardiology, vascular medicine, and radiotherapy departments could provide additional perspectives on cardiac surveillance.

However, with this study, we primarily aimed to explore the perceptions of oncology professionals because they are often the “gatekeepers” for the oncological care trajectories in the Netherlands. 

Oncologists or oncology nurse practitioners are the first professionals with whom a patient has contact once they are diagnosed with a malignancy. They provide a first draft of a patient’s treatment plan after which the patient is referred to a radiotherapist and surgeon, if indicated. 

Subsequently, with multidisciplinary discussion and further diagnostic procedures, the treatment plan is finalized. We believe that this would also be an ideal moment for oncologists to assess whether a patient has an increased cardiovascular risk based on their baseline characteristics and the scheduled treatments, and – if indicated – refer the patient to a cardiologist. 

Additionally, a previous study suggests that, in current practice, patients rarely receive any form of cardiac surveillance. Therefore, we were primarily interested in the perceptions of oncology professionals and potential influencing factors of delivering cardiac surveillance. 

Comment 2 reviewer 2

It is not clear why an epidemiologist was included in an interview about clinical practices.

Response

The epidemiologist included in our study is employed at a hospital specialized in oncological care; her work primarily focuses on improving breast cancer care and delivering cardiac surveillance in practice. This participant closely works with oncologists to improve current practice and has a wide knowledge of cardiac surveillance and its influencing factors. Therefore, she was deemed eligible for study participation.

---

## [Decision Letter · Decision Letter 1]

11 Mar 2021

Oncology professionals’ perspectives towards cardiac surveillance in breast cancer patients with high cardiotoxicity risk: A qualitative study

PONE-D-20-31818R1

Dear Dr. Koop,

We’re pleased to inform you that your manuscript has been judged scientifically suitable for publication and will be formally accepted for publication once it meets all outstanding technical requirements.

Kind regards,

Katriina Aalto-Setala, Professor

Academic Editor

PLOS ONE

Additional Editor Comments (optional):

Reviewers' comments:

Reviewer's Responses to Questions

**Comments to the Author**

1. If the authors have adequately addressed your comments raised in a previous round of review and you feel that this manuscript is now acceptable for publication, you may indicate that here to bypass the “Comments to the Author” section, enter your conflict of interest statement in the “Confidential to Editor” section, and submit your "Accept" recommendation.

Reviewer #1: All comments have been addressed

2. Is the manuscript technically sound, and do the data support the conclusions?

Reviewer #1: Yes

3. Has the statistical analysis been performed appropriately and rigorously? 

Reviewer #1: Yes

4. Have the authors made all data underlying the findings in their manuscript fully available?

Reviewer #1: Yes

5. Is the manuscript presented in an intelligible fashion and written in standard English?

Reviewer #1: Yes

6. Review Comments to the Author

Reviewer #1: The authors have addressed all concerns. Their responses are complete. No other concerns. Acceptable for publication.

7. PLOS authors have the option to publish the peer review history of their article (what does this mean?). If published, this will include your full peer review and any attached files.

Reviewer #1: No

---

## [Editor Report · Acceptance letter]

22 Mar 2021

PONE-D-20-31818R1 

Oncology professionals’ perspectives towards cardiac surveillance in breast cancer patients with high cardiotoxicity risk: A qualitative study 

Dear Dr. Koop:

I'm pleased to inform you that your manuscript has been deemed suitable for publication in PLOS ONE. Congratulations! Your manuscript is now with our production department. 

Kind regards, 

on behalf of

Dr Katriina Aalto-Setala 

Academic Editor

PLOS ONE